# Spectral Signatures of Hydrogen Thioperoxide (HOSH) and Hydrogen Persulfide (HSSH): Possible Molecular Sulfur Sinks in the Dense ISM

**DOI:** 10.3390/molecules27103200

**Published:** 2022-05-17

**Authors:** Charles Z. Palmer, Ryan C. Fortenberry, Joseph S. Francisco

**Affiliations:** 1Department of Chemistry & Biochemistry, University of Mississippi, University, MS 38677, USA; cpalmer5@go.olemiss.edu; 2Department of Earth and Environmental Sciences, University of Pennsylvania, Philadelphia, PA 19104, USA; frjoseph@sas.upenn.edu

**Keywords:** computational spectroscopy, astrochemistry, vibrational spectroscopy, anharmonic frequencies, rotational spectroscopy, quantum chemistry, coupled cluster theory

## Abstract

For decades, sulfur has remained underdetected in molecular form within the dense interstellar medium (ISM), and somewhere a molecular sulfur sink exists where it may be hiding. With the discovery of hydrogen peroxide (HOOH) in the ISM in 2011, a natural starting point may be found in sulfur-bearing analogs that are chemically similar to HOOH: hydrogen thioperoxide (HOSH) and hydrogen persulfide (HSSH). The present theoretical study couples the accuracy in the anharmonic fundamental vibrational frequencies from the explicitly correlated coupled cluster theory with the accurate rotational constants provided by canonical high-level coupled cluster theory to produce rovibrational spectra for use in the potential observation of HOSH and HSSH. The $\nu_{6}$ mode for HSSH at 886.1 cm${}^{-1}$ is within 0.2 cm${}^{-1}$ of the gas-phase experiment, and the $B_{0}$ rotational constant for HSSH of 6979.5 MHz is within 9.0 MHz of the experimental benchmarks, implying that the unknown spectral features (such as the first overtones and combination bands) provided herein are similarly accurate. Notably, a previous experimentally-attributed 2$\nu_{1}$ mode, at 7041.8 cm${}^{-1}$, has been reassigned to the $\nu_{1}+\nu_{5}$ combination band based on the present work’s $\nu_{1}+\nu_{5}$ value at 7034.3 cm${}^{-1}$. The most intense vibrational transitions for each molecule are the torsions, with HOSH having a more intense transition of 72 km/mol compared to HSSH’s intensity of 14 km/mol. Furthermore, HOSH has a larger net dipole moment of 1.60 D compared to HSSH’s 1.15 D. While HOSH may be the more likely candidate of the two for possible astronomical observation via vibrational spectroscopy due to the notable difference in their intensities, both HSSH and HOSH have large enough net dipole moments to be detectable by rotational spectroscopy to discover the role these molecules may have as possible molecular sulfur sinks in the dense ISM.

## 1. Introduction

Since the detection of sulfur-bearing molecules, such as carbon monosulfide [1,2], in the 1970s, in the interstellar medium (ISM), sulfur-containing species have captured the interest of astrochemists, astrophysicists, and astronomers for their analogous nature to oxygen and use in the determination of the physical structure of early stage star formation [3,4]. While volatile forms of sulfur molecules are known to exist in the diffuse ISM, the observed abundance of such molecules in the dense ISM was discovered to be less than 1% of the observed cosmic abundance [5]. This so-called sulfur depletion problem has led to the search for the missing sulfur in both the diffuse and dense regions of the ISM alike [6,7]. Since then, the majority of the sulfur that should be present in the dense ISM has still eluded detection. The majority of refractory forms of sulfur-bearing molecules have been hypothesized [8] to be depleted onto dust-grains found in the clouds of protoplanetary disks in some unknown molecule, but such a mechanism has not been supported or refuted. Alternatively, molecular forms of sulfur may simply exist in currently undetected molecular species in the ISM.

Studies of sulfur-bearing molecules in the dense ISM have led to the investigation of similar molecules in cometary ices [8] due to the hypothesis that these environments mimic the chemical nature of ice found in hot core formation. Recent astrochemical models support this hypothesis and predict sulfur-containing species most likely being bound to the surface of dust grains or in volatile ices [9]. This theory is further supported by the discovery of both solid H${}_{2}$S by the Rosetta orbiter in the Hale–Bopp comet [10] and carbonyl sulfide (OCS) in interstellar ice grain mantles of young stellar objects, such as AFGL 989 and Mon R2 IRS 2 [11]. These discoveries set the precedent for the detection of sulfur on dust grains in protoplanetary disks, but for which particular molecules to search is still unclear.

A natural starting point into the investigation of sulfur-containing species may stem from their analogous nature to oxygen-bearing molecules that have previously been detected. In 2011, HOOH was first discovered in the ISM and is hypothesized [12] to form through the addition of H atoms onto molecular oxygen on the surface of dust grains. A straightforward analog may occur from the addition of H atoms onto sulfur monoxide (SO), which was first observed in the ISM in 1973 [12,13]. Gaseous SO has been found in the remnants of type II supernovae that may contribute to the depletion of sulfur onto the dust grains found in the dense ISM [5,14]. If SO exists on the surface of dust grains, it may lead naturally to the formation of the sulfur analog hydrogen thioperoxide (HOSH). Hydrogen persulfide (HSSH) may form in a related way, but the polymeric allotropes of sulfur imply that it may be created under different conditions. In any case, observation of either molecule would provide in situ comparison for such mechanisms.

Consequently, the detection of HOSH and HSSH will require rovibrational reference data, which are provided herein via highly-accurate quantum chemical computations known as quartic force field(s) (QFFs). A QFF is a fourth-order Taylor series expansion of the potential portion of the internuclear molecular Hamiltonian [15]. QFFs have been used to produce highly-accurate rotational constants and fundamental anharmonic frequencies within to 1% of experimental values for numerous molecular systems [16,17,18,19,20,21,22,23,24,25,26,27,28,29]. The QFFs in this work are used in conjunction with high-level quantum chemical electronic structure computations and benchmarked with gas-phase experiment and previous theoretical studies where available to provide the necessary and full set of rovibrational spectral data needed for the potential detection of these sulfur-bearing molecules as possible reservoirs of sulfur in the dense region of the ISM.

## 2. Computational Details

The present work utilizes two different methods of producing QFFs for each molecule. The two methods are based on coupled cluster theory at the singles, doubles, and perturbative triples [CCSD(T)] level [30]. The first method implements the CCSD(T) level of theory within the explicitly correlated F-12b formalism along with the cc-pVTZ-F12 basis set [31,32]. This method will henceforth be abbreviated as F12-TZ. The second method is a composite method based on canonical CCSD(T) that takes into account effects from the complete basis set (CBS) limit extrapolation (“C”), core electron correlation (“cC”), and scalar relativity (“R”) yielding the abbreviated “CcCR” method. F12-TZ QFFs have been used to compute accurate anharmonic fundamental frequencies that are as, if not more, accurate than the frequencies computed using the CcCR method but at a fraction of the computational cost [33,34,35]. Even though F12-TZ provides highly accurate anharmonic fundamental frequencies, it cannot readily produce rotational constants at the same level of accuracy as the CcCR method, motivating the usage of both QFF methods herein [29].

Both QFF methods utilize the MOLPRO 2020.1 quantum chemical package [36]. Both methods begin with the optimization of the molecular geometry with exceptionally tight convergence criteria. The CcCR method utilizes a geometry optimized at the CCSD(T)/aug-cc-pV5Z level of theory, but includes corrections to this geometry based on the difference between the inclusion and exclusion of the effects of core electron correlation from the Martin–Taylor [37] (MT) core-correlating basis set. The F12-TZ method, however, employs a simpler approach by only optimizing at the CCSD(T)-F12b/cc-pVTZ-F12 level of theory. Once each geometry has been optimized, the structures of each species are displaced by 0.005 Å for bond lengths and 0.005 radians for bond angles/torsions using symmetry–internal coordinates via the INTDER [38] program.

The simple–internal coordinate system for HOOH requires 665 points for the QFF and is defined from the atoms in Figure 1a [39,40]:(1)$$\begin{array}{cc} S_{1} & =r(O_{2}-O_{3}) \end{array}$$(2)$$\begin{array}{cc} S_{2} & =\frac{1}{\sqrt{2}}\left[r\left(H_{1}-O_{2}\right)+r\left(O_{3}-H_{4}\right)\right] \end{array}$$(3)$$\begin{array}{cc} S_{3} & =\frac{1}{\sqrt{2}}\left[∠\left(H_{1}-O_{2}-O_{3}\right)+∠\left(O_{2}-O_{3}-H_{4}\right)\right] \end{array}$$(4)$$\begin{array}{cc} S_{4} & =\tau (H_{1}-O_{2}-O_{3}-H_{4}) \end{array}$$(5)$$\begin{array}{cc} S_{5} & =\frac{1}{\sqrt{2}}\left[r\left(H_{1}-O_{2}\right)-r\left(O_{3}-H_{4}\right)\right] \end{array}$$(6)$$\begin{array}{cc} S_{6} & =\frac{1}{\sqrt{2}}\left[∠\left(H_{1}-O_{2}-O_{3}\right)-∠\left(O_{2}-O_{3}-H_{4}\right)\right] \end{array}$$

The simple–internal coordinate system for HSSH is the same as the HOOH system with S atoms replacing the O atoms. The symmetry–internal system for HOSH requires 1289 points for the QFF and is based on the geometry of Figure 1b:(7)$$\begin{array}{cc} S_{1} & =r(H_{1}-O) \end{array}$$(8)$$\begin{array}{cc} S_{2} & =r(O-S) \end{array}$$(9)$$\begin{array}{cc} S_{3} & =r(S-H_{2}) \end{array}$$(10)$$\begin{array}{cc} S_{4} & =∠(H_{1}-O-S) \end{array}$$(11)$$\begin{array}{cc} S_{5} & =∠(O-S-H_{2}) \end{array}$$(12)$$\begin{array}{cc} S_{6} & =\tau (H_{1}-S-O-H_{2}) \end{array}$$

A single-point energy calculation for every displacement for either QFF method is then computed. At each displaced geometry point using the CcCR method, CCSD(T) energies are computed with the aug-cc-pV(T+d)Z, aug-cc-pV(Q+d)Z, and aug-cc-pV(5+d)Z basis sets for the extrapolation to the CBS limit [41]. Each point is also computed with the core correlation turned on and off utilizing the same MT basis set from the geometry optimization. Additionally, the Douglas–Kroll scalar relativistic corrections [42] are computed using the cc-pVTZ-DK basis set with the corrections turned either on or off. Finally, single-point energy calculations at each displaced geometry for HOSH only were computed using the CCSDT/aug-cc-pVTZ level of theory for the inclusion of the contributions of full triples in a separate, but similar, “CcCRE” composite method. For F12-TZ, each single-point energy calculation is only computed at the CCSD(T)-F12b/cc-pVTZ-F12 level of theory. Regardless of QFF method, once the single-point energy calculations are finished, the relative energies are fit to the QFF Taylor series model using a least squares fit procedure to better than 10${}^{-17}$ a.u.${}^{2}$, then fit once again to include contributions from the computed Hessian to construct the equilibrium geometry. The force constants generated from the least squares procedure are re-fit to produce zero gradients and then are transformed into Cartesian coordinates through the INTDER program [38]. These are then used by the SPECTRO [43] program to compute the spectroscopic constants and vibrational frequencies produced by rotational and vibrational perturbation theory at second order (VPT2) [44,45,46].

The rovibrational spectra of each species contains Fermi resonances and resonance polyads. These are treated by the SPECTRO program to provide more accurate predictions [47] of the rovibrational spectra. The spectrum for HOSH contains a 2$\nu_{6}=\nu_{4}$ and a 2$\nu_{6}=\nu_{5}$ type-1 Fermi resonance. For HSSH’s spectrum, a 2$\nu_{5}=\nu_{6}$ and a 2$\nu_{5}=\nu_{3}$ type-1 Fermi resonance and $\nu_{2}/\nu_{1}$, $\nu_{2}/\nu_{1}$, and $\nu_{2}/\nu_{1}$ Darling–Dennison resonances are present, and the spectrum for HOOH includes a $\nu_{6}=\nu_{5}$ type-1 Fermi resonance, a $\nu_{6}+\nu_{5}$ = $\nu_{3}$ type-2 Fermi resonance, and a $\nu_{2}/\nu_{1}$ and $\nu_{4}/\nu_{3}$ Darling–Dennison resonance. To further assist in possible observation, dipole moments for each species are computed at the CCSD(T)-F12b/cc-pVTZ-F12 level of theory. Anharmonic infrared intensities are calculated using the Gaussian16 [48] quantum chemical package at the MP2/aug-cc-pVDZ level of theory, which has been shown to produce a semi-quantitative agreement with higher levels of theory for far less computational costs [49,50].

## 3. Results and Discussion

### 3.1. HOOH

The computed rovibrational spectra for HOOH in this work provide reference benchmarks to show spectral differences between it and the sulfur analogs. As seen in Table 1, F12-TZ surprisingly outperforms CcCR by producing the more accurate rotational constants compared to experiment. F12-TZ’s $B_{0}$ value of 26,171.3 MHz is lower than the experimental [51] value of 26,194.08965 MHz by 22.8 MHz, an error of 0.09%. In contrast, the $B_{0}$ rotational constant computed via the CcCR QFF method is 82.1 MHz higher than the experimental value with the $C_{0}$ following suit, having a difference of 169.2 MHz. This, however, is not surprising, as previous computational studies [52] on this molecule also fail to capture accurate rotational constants possibly due to the large amplitude motion of the torsion. Regardless, the difference between CcCR and the experiment for the $B_{0}$ and $C_{0}$ constants are 0.31% and 0.67% in error, respectively.

With regard to most vibrational frequencies, the present work agrees well with both the previous theory and experiment, as shown in Table 2. The fundamental frequencies produced by the F12-TZ method compares well with the gas-phase experiment [57] with the $\nu_{2}$ frequency, the symmetric H–O–O bend at 1393.9 cm${}^{-1}$, being 0.4 cm${}^{-1}$ higher than the experimental fundamental of 1393.5 cm${}^{-1}$. CcCR compares similarly with the $\nu_{3}$ frequency, the O–O stretch with a fundamental of 878.4 cm${}^{-1}$, being 0.5 cm${}^{-1}$ higher than gas-phase value of 877.93 cm${}^{-1}$. The worst agreement is the $\nu_{4}$ frequency, the torsion, with F12-TZ and CcCR being 51.5 and 56.2 cm${}^{-1}$ lower than experiment, respectively. Once again, as discussed in previous literature [52], this may be attributed to the large amplitude motion of the torsion. A previous theoretical study [52] utilizes a similar QFF method also employing the CCSD(T)-F12/cc-pVTZ-F12 level of theory, but the previous work uses a different fitting model than the present work and does not mention the inclusion of resonance polyads in the VPT2 corrections. To that end, the difference between the previous theoretical study and the present work is to be expected with most fundamental frequencies being within 0.5 cm${}^{-1}$, and the worst agreement being the $\nu_{6}$ fundamental with less than an 8.0 cm${}^{-1}$ difference.

The inaccuracies in both the rotational constants and the anharmonic vibrational frequencies for $\nu_{4}$ warrant an investigation of the potential well for this torsional motion of HOOH. As seen in Figure 2A, a relaxed scan of the torsional angle produces an extremely flat potential well with a torsional trans-barrier of 371.9 cm${}^{-1}$ (1.06 kcal/mol). A previous theoretical study [60] also investigated the trans-barrier height, seen in Figure 2A, and calculates the trans-barrier 1.17 kcal/mol higher than the current work at 2.23 kcal/mol (∼780 cm${}^{-1}$). However, a previous experimental value [61] for the trans-barrier at 387.07 cm${}^{-1}$ is closer to the present work’s trans-barrier than previous theory. The previous theory utilizes the HF/STO level of theory so the inaccuracy compared to the experiment is to be expected. In any case, a molecule with a mode that exhibits a flat potential can be notorious for its inability to be accurately modeled with VPT2 corrections as used in the present work. As stated previously, CcCR QFF methods generally provide more accurate ground state rotational constants. However, due to the composite nature of the method, there are conflicting minima for the potential energy surface, thus introducing additional inaccuracy. This same inaccuracy has been seen as well for [Al, N, C, O] isomers [62]. While there is still considerable accuracy to experiment for the remainder of the fundamentals, the large amplitude motion of the torsion decreases the overall accuracy of the ground state rotational constants, seemingly preventing the accurate rovibrational modeling of this molecule.

Further, several two-quanta vibrational overtones and combination bands for HOOH have not been reported in the current literature. In order to assist in further potential astrophysical observation, the present work introduces the missing overtones and combination bands for HOOH, as seen in Table 3. In current literature, two previous studies attribute the same value of 7041.8 cm${}^{-1}$ to either the 2$\nu_{1}$ or $\nu_{1}+\nu_{5}$ two-quanta band, Halonen [63] and Redington et al. [64], respectively. The present work’s F12-TZ value for the $\nu_{1}+\nu_{5}$ combination band is 7034.3 cm${}^{-1}$ falling in a similar region with the value attributed to the value from Redington et al. [64] with only a difference of 7.5 cm${}^{-1}$. Another previous experimental value, from Dzugan et al. [65], for the $\nu_{1}+\nu_{5}$ combination band, at 7050 cm${}^{-1}$, also falls within the same region as the present work’s theoretical value along the work from both Halonen [63] and Redington et al. [64]. Based on the agreement between the present work’s value and each of the previous studies values for the band in question, this band is attributed to the $\nu_{1}+\nu_{5}$ combination band for HOOH.

With regard to other two-quanta modes, previous gas-phase experimental work is available for comparison [64]. The previous experimental $\nu_{1}+\nu_{6}$ combination band, at 4827.49, does not compare favorably to the present F12-TZ value of 4873.8, or a 46.3 cm${}^{-1}$ difference. Both the $\nu_{2}+\nu_{5}$ and $\nu_{3}+\nu_{5}$ two-quanta bands from the previous gas-phase experiment [64], however, compare exceptionally well to the present theoretical study. The previous gas-phase $\nu_{2}+\nu_{5}$ value of 4982.57 cm${}^{-1}$ is 0.8 cm${}^{-1}$ higher than the present F12-TZ combination band. Similarly, the previous gas-phase $\nu_{3}+\nu_{5}$ value of 4487.27 cm${}^{-1}$ is 2.4 cm${}^{-1}$ higher than the present CcCR value of 4484.9 cm${}^{-1}$. While some two-quanta modes compare well to experiment, the slight inaccuracy compared to the experiment for the other two-quanta bands is not unexpected as the available previous gas-phase experiment was conducted in 1962. Therefore, the present work’s theoretical values will serve as a benchmark for any further laboratory analysis, even for the overtones and combination bands.

Analysis of the computed two-quanta mode intensities show far weaker transitions than the anharmonic vibrational frequencies with the brightest two transitions, the $\nu_{1}+\nu_{5}$ and $\nu_{2}+\nu_{6}$ combination bands, at a mere 5.0 km/mol. Outside of these modes, no two-quanta modes are present that exhibit intensities greater than 1.0 km/mol. Though these transitions are weak, the $\nu_{1}+\nu_{5}$ and $\nu_{2}+\nu_{6}$ band fall within the elusive, and consequently understudied, near- to mid-IR spectrum, 1.4 μm and 3.7 μm range, respectively. With the recently launched James Webb Space Telescope (JWST), the proper instrumentation to analyze this region of the IR spectrum is now achievable with its near infrared spectrograph. Since these fall within the region of the IR dominated by polycyclic aromatic hydrocarbons (PAHs), the data provided in this work will be instrumental in identifying molecules in this region that are unrelated to these PAHs but have yet to be identified.

### 3.2. HOSH

Considering HOSH’s analogous nature and similar geometry to HOOH, a potential energy scan of the torsional motion for this molecule is also investigated. In Figure 2B, a considerably deeper potential well, with a trans-barrier height of 1536.4 cm${}^{-1}$ for the torsion, is seen compared to the potential well for HOOH. With this deeper well, the rovibrational spectra of HOSH will not suffer from the same inaccuracies in its VPT2 corrections. For this reason, the geometrical parameters and rotational constants, given in Table 4, show a much higher accuracy compared to experiment than that of HOOH. The CcCR value for $B_{0}$ of 15,299.9 MHz is in good agreement with the experimental gas-phase value of 15,282 MHz [67], giving a difference of only 17.9 MHz, which is only an error of 0.12% [29]. The F12-TZ value is less accurate, as expected, with a difference of ∼40 MHz, an error of 0.26%, further supporting the accuracy of rotational constants computed using CcCR versus F12-TZ when compared to the experiment. Similarly, the $C_{0}$ rotational constant shares a 20.4 MHz difference, 0.14% error, between the CcCR value and the previous experimental value of 14,840 MHz, while the F12-TZ $C_{0}$ value is 34.7 MHz lower than the experimental method, an error of 0.23%. A previous theoretical study [68], however, performs more accurately when compared to the experiment and calculates rotational constants at the CCSD(T, full)/cc-pwCVQZ level with vibration–rotation corrections from the CCSD(T)/cc-pV(T+d)Z level of theory and finds the $B_{0}$ and $C_{0}$ to be within 7.0 and 3.0 MHz, respectively. The difference in accuracy for these rotational constants may largely be due to the previous computational study’s implementation of the CCSD(T, full)/cc-pwCVQZ level of theory with vibrational corrections at the CCSD(T)/cc-pV(T+d)Z level, which is considered to be a more theoretically rigorous composite method.

Additionally, a previous theoretical study [71] simulates the full rotational line spectrum of HOSH at the CCSD(T)/aug-cc-pV(Q+d) level of theory through the use of the TROVE program [72]. While the present study utilizes VPT2 to generate accurate rovibrational constants and fundamental frequencies, TROVE implements a variational method for generating accurate rotational energies. The present CcCR $B_{0}$ rotational constant of 15,299.9 MHz is 16.1 MHz higher than the previous theory’s $B_{0}$ of 15,283.8 MHz, which is only in error of 0.11%. The comparable ground state rotational constants are derived from the rotational energies provided in the previous study. Furthermore, the CcCR $C_{0}$ value of 14,860.4 MHz is 22.1 MHz greater than the previous theoretical $C_{0}$ value of 14,838.3, only a 0.15% error. While the previous theoretical ground state rotational constants are more accurate compared to experiment, the small margin of error between the present and previous methods still suggests the validity and accuracy of the current VPT2 methodology for generating accurate rotational constants for systems of this type.

Presently, HOSH only has two observed fundamental frequencies by previous gas-phase experiment [73], as seen in Table 5: the O–H stretch at 3625.6 cm${}^{-1}$ and the, tentatively assigned, S–H stretch at 2538 cm${}^{-1}$. The present F12-TZ $\nu_{1}$ fundamental of 3626.7 cm${}^{-1}$ compares favorably with the gas-phase value of 3625.6 cm${}^{-1}$. Both the present work and previous theory compare similarly with the gas-phase $\nu_{2}$ fundamental at 2538 cm${}^{-1}$ with the F12-TZ fundamental being 6.9 cm${}^{-1}$ higher than experiment, and previous theory at 4.7 cm${}^{-1}$ lower than the experimental value.

In a previous experimental work conducted by Beckers et al. [73], the IR spectrum of the S–H stretch was investigated and shown to be just above the $\nu_{1}+\nu_{3}$ combination band from the SO${}_{2}$ byproduct from the experiment. Due to the overlap from this combination band on the lower J branches, the previous work was unable to confirm the full assignment of this fundamental mode. In 2009, Yurchenko et al. [74] simulated an IR spectrum for the S–H stretching region, utilizing the TROVE program at the aug-cc-pV(Q+d)Z level of theory, to explain perturbations found in the experimental S–H stretching frequency. The previous simulated spectrum is in good qualitative agreement with the region around the band center of the S–H stretching region, but lacks any comparison to the lower or higher frequency bands. The present study provides a simulated IR spectrum generated through PGOPHER’s [75] vibrational spectrum simulation software. In Figure 3, the S–H stretching fundamental frequency is centered on the previous experiment’s fundamental at 2538 cm${}^{-1}$ in order to compare the overall rovibrational structure from the presently-computed vibrationally excited rotational constants provided by the use of the VPT2 methodology in this work with that from the previously reported laboratory spectrum. The bands align closely with the lower J bands of the previous experimental IR spectrum, only deviating after the third band. These deviations from higher frequencies can be attributed to the present work’s rotational constants being lower than experiment, which is expected.

Moving toward detectability, the anharmonic intensities, seen in Table 5, show two relatively high intensity vibrational transitions. HOSH’s most intense transition is, again, the $\nu_{6}$ torsion at 72 km/mol, while the second is the $\nu_{1}$, O–H stretch, fundamental transition at 67 km/mol. Compared to what is considered the intense transition of the anti-symmetric stretch of water at 70 km/mol, the two aforementioned transitions of HOSH are of similar intensity suggesting these are readily detectable. An additional benefit of the use of the QFF methods in this work is their ability to produce not only highly accurate ground vibrational state rotational constants, but also vibrationally excited rotational constants. The present work introduces such rotational constants for multiple fundamentals that serve to assist in accurate rovibrational modeling of HOSH as is discussed above for $\nu_{2}$. These models are constructed to support further laboratory analysis or potential astronomical observation through the use of vibrationally excited rotational spectroscopy as is the case for the detection of vibrationally excited states of the SiS [76] and C${}_{6}$H [77] molecules observed in IRC+10216.

Similar to HOOH, the current literature discussing the two-quanta vibrational overtones and combination bands for HOSH is limited. To aid in this regard, the present work introduces such two-quanta modes for the purpose of benchmarking, assistance in potential astrophysical detection, and further laboratory analysis, as seen in in Table 6. Comparing to previous literature, a previous theoretical value [74] for the first overtone of the torsional motion, 2$\nu_{6}=$ 846.269 cm${}^{-1}$, is only 7.2 cm${}^{-1}$ higher the the present work’s CcCR value of 839.1 cm${}^{-1}$. This further suggests the reliability of the present theoretical methodology for generating two-quanta bands for molecules of this type. With regard to the detectability of these two-quanta transitions, much like in HOOH, the relative intensities are exceptionally lower than that of the anharmonic vibrational frequencies. The brightest transitions are the first overtone of both the O–H stretch and the O–S–H bend at 5 km/mol and 4 km/mol, respectively. Once again, these two-quanta modes fall within the near- to short-wavelength mid-IR spectrum that the JWST will be able to probe more efficiently. Thus, the present work provides the necessary benchmark data for the potential astrophysical detection of HOSH via investigation of its overtones and combination bands.

### 3.3. HSSH

For much of the same reason as HOSH, a potential energy scan of the torsion for HSSH is investigated to probe viability with the current QFF methods. In Figure 2C, an even deeper potential well for this motion is shown compared to both HOSH and HOOH. This comparison can be clearly seen in Figure 2C where the torsional PES scans of all three molecules are given. The trans-barrier height of 2026.6 cm${}^{-1}$is the highest seen for each of the three molecules investigated in this work and is considerably higher than the HOOH trans-barrier height of 371.9 cm${}^{-1}$. That being said, as with HOSH, the current rovibrational spectrum for this molecule should not suffer from the inaccuracies in the VPT2 corrections that are a byproduct of the shallow potential well.

With regard to the rotational constants of HSSH, provided in Table 7, considerable agreement is demonstrated in the present study’s F12-TZ $B_{0}$ rotational constant of 6979.5 MHz being only 9.0 MHz above the gas-phase value, an error of 0.13% [78]. The same agreement is not present for the CcCR method’s value of 7010.6 MHz being nearly 40.0 MHz above previous gas-phase experiment; nevertheless this difference is only 0.58% in error. With regard to the $C_{0}$ rotational constant, the CcCR method produces a value of 6938.1 MHz which is only 30.0 MHz below the 6967.68832 MHz gas-phase constant. The F12-TZ $C_{0}$ does not fare similarly, however, with its computed value being nearly 74.0 MHz lower than that from gas-phase experiment. As expected, the CcCR method produces a more accurate $C_{0}$ value only in error of 0.42%, while the F12-TZ value is in error of roughly 1.1%. Additionally, the current QFF methodology provides the computation of the vibrationally excited rotational constants that have not been previously investigated possibly offering Appendix A that may be necessary in the potential detection of this molecule rotationally or rovibrationally in the infrared.

Shown in Table 8, exceptional agreement can be seen between the two present QFF methods for the anharmonic vibrational frequencies of HSSH. The S−S stretch, fundamental $\nu_{3}$, shows the best agreement between the computational values with F12-TZ’s value of 516.8 cm${}^{-1}$ being within 2.0 cm${}^{-1}$ of CcCR. All fundamentals for HSSH are within 4.0 cm${}^{-1}$ between QFF methods, except for the CcCR torsional motion of $\nu_{4}$ being 11.0 cm${}^{-1}$ below that of the F12-TZ method. When comparing to experiment, F12-TZ performs better with the best agreement being the H–S–S anti-symmetric bend, $\nu_{6}$, at 886.1 cm${}^{-1}$, being less than 0.2 cm${}^{-1}$ above previous gas-phase experiment $\nu_{3}$ [85]. There is also considerable agreement between the previous gas-phase experiment’s S−S stretching frequency and that from F12-TZ being within 0.9 cm${}^{-1}$ of one another. The same agreement with the experiment is not suggested with regard to CcCR as almost every mode differs by more than 10.0 cm${}^{-1}$, save for the $\nu_{6}$ mode with a difference of 2.0 cm${}^{-1}$. The difference between the CcCR QFF and experiment re-illustrates the high accuracy of the F12-TZ QFF method for anharmonic fundamental vibrational frequencies molecules of this type.

The anharmonic intensities of the vibrational transitions for HSSH are reported in Table 8. While the highest intensity transition, such as HOSH, is the torsion, HSSH has by far the lowest intense of the molecules investigated in this work. HSSH’s torsional motion has a calculated intensity of 14 km/mol, while it is second highest intensity transition is the $\nu_{6}$ S−S–H bend at 2.0 km/mol. These are glaringly less intense than HOSH’s 72 km/mol intensity for its $\nu_{6}$, torsional, transition suggesting that HSSH is not nearly as observable via IR spectroscopy. Both the gas-phase experiment or theoretical study appear to be lacking with regard to the dipole moment. For this reason, while the present computed dipole moment of HSSH (1.15 D) is relatively small compared to the other sulfur-analog investigated in this work, it should nonetheless serve as a basis for potential radio-astronomical observation.

Furthermore, the present work introduces calculated two-quanta vibrational overtones and combination bands that appear to be missing from the current literature, as seen in Table 9. There is previous experimental work [78] for the first overtone of the torsional motion, 2$\nu_{4}=$ 808.0 cm${}^{-1}$, that is only 3.8 cm${}^{-1}$ lower than the present work’s F12-TZ value. Unlike HOOH and HOSH, HSSH exhibits no overtone or combination band transitions that have intensities over 1.0 km/mol potentially reducing its chance of detection through the use of IR spectroscopy and explaining why sulfur-containing molecules may be underdetected in astrophysical sources. Nonetheless, the present work’s introduction of the two-quanta modes provides reference data for further laboratory benchmarking or potential astrophysical detection.

## 4. Conclusions

HOSH is the more likely candidate for potential astronomical detection as a possible sulfur sink in the dense ISM, compared to HSSH. While the vibrational spectrum for HSSH contains more experimentally observed frequencies, the calculated intensities of those transitions are markedly lower than that of HOSH, including the first overtones and combination bands. Each molecule investigated in this work shows the most intense and lowest energy anharmonic vibrational frequency to be the torsional mode. For the investigation of the two sulfur-bearing analogs, the computed anharmonic intensities and dipole moments presented in this work show that the more intense torsional transition belongs to HOSH at 72 km/mol, which noticeably outshines HSSH’s torsional transition of a mere 14 km/mol. Additionally, HOSH’s net dipole moment of 1.60 D is almost 0.5 D greater than HSSH’s 1.15 D, further supporting HOSH as the more readily detectable sulfur-bearing molecule, both rotationally and in the infrared.

Additionally, the present work has investigated the potential energy wells of the torsional motion for HOOH, HOSH, and HSSH. HOOH exhibits a relatively flat potential that diminishes the accuracy of the torsional fundamental as well as the ground state rotational constants when compared to both the experiment and previous theory. The potential energy scans for HOSH and HSSH, however, show much deeper potential wells for this torsional motion, and the rovibrational spectra indicate that these molecules do not suffer from the same inaccuracies in their fundamentals and rotational constants. To that end, the present work includes not only ground state rotational constants, but also introduces vibrationally excited rotational constants for numerous fundamentals for both HOSH and HSSH that should exhibit the same level of accuracy. Additionally, the present work includes novel two-quanta modes for HOOH that have yet to be reported in the literature and are included for the first time. This accuracy for the vibrationally excited rotational constants is demonstrated for HOSH through their use alongside a previous experimental IR spectrum for the S–H stretching fundamental [73] centered at 2538 cm${}^{-1}$ simulating an IR spectrum that especially aligns well with the previous gas-phase experiment for the lower J values. This semi-quantitative agreement from the use of the vibrationally excited rotational constants provided in this work suggests that the rovibrational spectra for HOSH produced via the present QFF methodology can be trusted to serve as a basis and a benchmark for any future comparison to—or assignments from—gas-phase IR spectroscopy.

The anharmonic fundamental vibrational frequencies, and their intensities, will be particularly useful for current observatories, such as the stratospheric observatory for infrared astronomy. With the torsional frequency being the most intense and the lowest energy transition, the inclusion of the vibrationally excited rotational constants will be beneficial for detection from spaced-based observatories that will have the capability for both high-resolution and high sensitivity spectroscopy, notably the recently launched JWST. Similarly, the two-quanta modes introduced in the present work provide overtones and combination bands for each molecule investigated in this work, and have been utilized to confirm the assignment of the precious gas-phase value of 7041.8 cm${}^{-1}$ to the $\nu_{1}+\nu_{5}$ combination band. The provided first overtones and combination bands fall in regions of the IR spectrum that have been understudied but can now be efficiently probed through the use of JWST, most notably its NIRSpec instrument. In this regard, the present work provides the necessary reference data to assist in the astrophysical detection of HOSH and perhaps HSSH. Finally, the rotational spectroscopic constants provided in this work will be essential for ground-based radio telescopic observation from facilities such as the Atacama Large Millimeter/submillimeter Array. Consequently, the investigation of HOSH and HSSH may provide a clue as to where the molecular sulfur in the dense ISM has been eluding detection for the past 50 years.

## Figures and Tables

**Figure 1 molecules-27-03200-f001:**
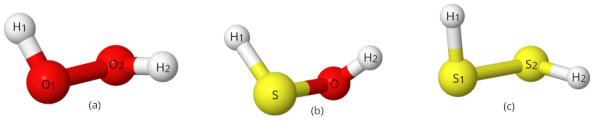
Visual depiction of the optimized structures for (**a**) HOOH, (**b**) HOSH, (**c**) HSSH.

**Figure 2 molecules-27-03200-f002:**
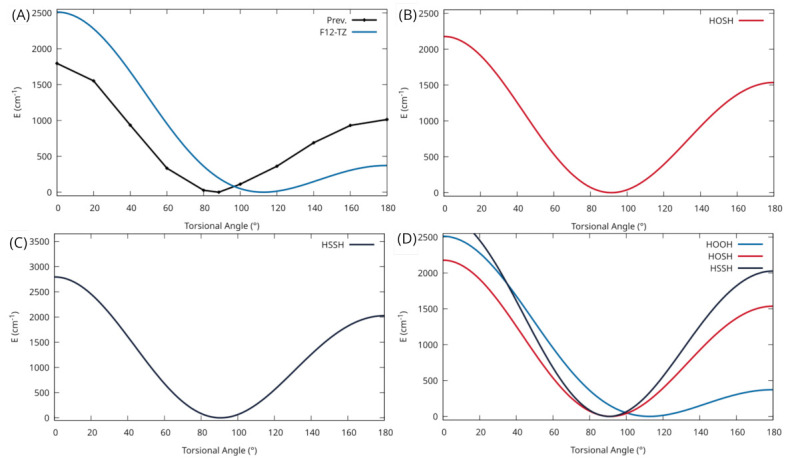
Potential energy scan of the torsional motion. (**A**) HOOH calculated at the F12-TZ level of theory (blue) and a previous theoretical study using HF/STO (black), (**B**) HOSH calculated at the F12-TZ level of theory, (**C**) HSSH at the F12-TZ level of theory, (**D**) comparison of HOOH, HOSH, and HSSH.

**Figure 3 molecules-27-03200-f003:**
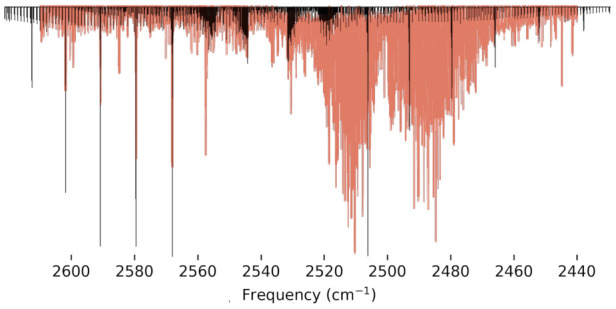
Simulated IR spectrum (black) of the S–H stretch for HOSH with the gas-phase IR spectrum from [73] (red) of the same mode containing the contamination from the ($\nu_{1}+\nu_{3}$) combination bands.

**Table 1 molecules-27-03200-t001:** Geometrical parameters and spectroscopic constants for HOOH compared to the previous gas-phase experiment.

	Units	CcCR	F12-TZ	Prev. Theory	Prev. Expt.
R${}_{e}$(H-O)	Å	0.96198	0.96348	0.963 ${}^{b}$	0.9617 ${}^{b}$
R${}_{e}$(O-O)	Å	1.44803	1.45099	1.450 ${}^{b}$	1.4524 ${}^{b}$
*∠*${}_{e}$(O-O-H)	${}^{\circ }$	100.15	100.09	100.1 ${}^{b}$	99.76 ${}^{b}$
τ${}_{e}$(H-O-O-H)	${}^{\circ }$	112.80	112.66	112.7 ${}^{b}$	113.6 ${}^{b}$
$A_{e}$	MHz	304,899.4	303,773.7		
$B_{e}$	MHz	26,576.3	26,472.3		
$C_{e}$	MHz	25,742.3	25,646.3		
R${}_{0}$(H${}_{1}$-O${}_{1}$)	Å	0.96228	0.96379	0.9675 ${}^{c}$	
R${}_{0}$(O${}_{1}$-O${}_{2}$)	Å	1.46166	1.46464	1.461 ${}^{c}$	
*∠*${}_{0}$(H${}_{1}$-O${}_{1}$-O${}_{2}$)	${}^{\circ }$	100.05	99.99	100.07 ${}^{c}$	
$A_{0}$	MHz	302,188.9	301,077.7	300,419 ${}^{b}$	301,874.2654 ${}^{a}$
$B_{0}$	MHz	26,276.2	26,171.3	26,030 ${}^{b}$	26,194.08965 ${}^{a}$
$C_{0}$	MHz	25,286.1	25,194.1	25,349 ${}^{b}$	25,116.88435 ${}^{a}$
$A_{1}$	MHz	297,349.8	296,967.2		
$B_{1}$	MHz	26,232.4	26,128.6		
$C_{1}$	MHz	25,282.6	25,191.3		
$A_{2}$	MHz	305,755.8	304,615.1		
$B_{2}$	MHz	26,093.7	25,988.4		
$C_{2}$	MHz	25,214.1	25,121.0		
$A_{3}$	MHz	301,557.4	300,454.1		301,873.4973264 ${}^{f}$
$B_{3}$	MHz	25,957.4	25,853.2		26,193.0529267 ${}^{f}$
$C_{3}$	MHz	24,962.0	24,871.3		25,117.8952429 ${}^{f}$
$A_{4}$	MHz	300,894.7	299,793.5		
$B_{4}$	MHz	26,290.4	26,182.9		
$C_{4}$	MHz	25,044.4	24,958.2		
$A_{5}$	MHz	295,700.7	293,937.2		
$B_{5}$	MHz	26,240.8	26,136.4		
$C_{5}$	MHz	25,284.4	25,193.1		
$A_{6}$	MHz	306,454.1	305,307.4		306,559.6731544 *^e^*
$B_{6}$	MHz	26,215.6	26,110.0		26,162.8878096 *^e^*
$C_{6}$	MHz	25,043.1	24,951.8		24,836.1562867 *^e^*
$\Delta_{J}$	kHz	96.817	96.409		86.6100411 *^e^*
$\Delta_{K}$	MHz	11.027	10.914		13.7784613 *^e^*
$\Delta_{JK}$	MHz	1.138	1.130		1.2576293 *^e^*
$\delta_{J}$	Hz	−31.906	−14.235		
$\delta_{K}$	MHz	6.812	6.780		
$\Phi_{J}$	mHz	392.379 (μHz)	1.285		
$\Phi_{K}$	kHz	1.765	1.729		
$\Phi_{JK}$	Hz	77.849	74.877		
$\Phi_{KJ}$	Hz	−228.455	−219.330		
$\phi_{j}$	μHz	2.608 (mHz)	887.135		
$\phi_{jk}$	Hz	111.163	111.449		
$\phi_{k}$	kHz	−53.653	−55.097		
μ	D		1.75	1.746 ${}^{d}$	

*^a^* Experimental ground state rotational constants from [51]. *^b^* Previous theory computed at CCSD(T)-F12/ccpVTZ-F12 from [52]. *^c^* Experimental geometrical parameters from [53]. *^d^* Electric dipole moment calculated at the CCSD(T)/aug-cc-pVTZ level of theory from [54]. *^e^* Experimental ν6 rotational constant for *ν*_6_ from [55]. *^f^* Experimental rotational constants for *ν*_6_ from [56].

**Table 2 molecules-27-03200-t002:** Vibrational frequencies (cm${}^{-1}$), and IR intensities (km/mol) given in parentheses for HOOH compared to the previous theory and gas-phase experiment.

Mode	Desc.	CcCR	F12-TZ	Prev. Theory ${}^{a}$	Prev. Theory ${}^{b}$	Prev. Expt.
$\omega_{1}$(*a*)	S${}_{5}$	3806.1	3798.5	3798		
$\omega_{2}$(*a*)	S${}_{3}$	1441.7	1437.6	1447		
$\omega_{3}$(*a*)	S${}_{1}$	915.0	911.3	911		
$\omega_{4}$(*a*)	S${}_{4}$	380.3	380.4	378		
$\omega_{5}$(*b*)	S${}_{2}$	3805.8	3798.4	3798		
$\omega_{6}$(*b*)	S${}_{6}$	1333.3	1330.1	1330		
$\nu_{1}$(*a*)	S${}_{5}$	3611.1	3607.5 (11)	3607	3611.05	3609.8 ${}^{c}$
$\nu_{2}$(*a*)	S${}_{3}$	1398.0	1393.9 (1)	1400	1394.43	1393.5 ${}^{d}$
$\nu_{3}$(*a*)	S${}_{1}$	878.4	875.3 (1)	875	865.77	877.93 ${}^{d}$
$\nu_{4}$(*a*)	S${}_{4}$	314.7	319.4 (164)	315		370.89 ${}^{c}$
$\nu_{5}$(*b*)	S${}_{2}$	3608.8	3605.2 (49)	3605	3609.73	3610.66 ${}^{c}$
$\nu_{6}$(*b*)	S${}_{6}$	1278.4	1275.9 (119)	1283	1264.78	1273.68 ${}^{f}$
ZPT		5740.5	5730.6			

*^a^* Previous theoretical QFF computed at CCSD(T)-F12/cc-pVTZ-F12 level of theory from [52]. *^b^* Previous theoretical values using VMP2 from [53]. *^c^* Previous gas-phase experimental values gathered from [56], *^d^* [57]. *^e^* [58], and *^f^* [59].

**Table 3 molecules-27-03200-t003:** Vibrational frequencies (cm${}^{-1}$) and IR intensities (km/mol) given in parentheses for two-quanta bands of HOOH compared to previous gas-phase experiment.

Mode	CcCR	F12-TZ	Prev. Expt.	Prev. Theory
$2\nu_{1}$	7131.0	7125.6 (1)		7041.8 ${}^{a}$
$2\nu_{2}$	2772.9	2764.8 (1)		
$2\nu_{3}$	1739.6	1734.0 (1)		
$2\nu_{4}$	557.8	571.5 (1)		
$2\nu_{5}$	7125.2	7119.8 (1)		
$2\nu_{6}$	2534.8	2529.7 (1)		
$\nu_{1}$ + $\nu_{2}$	4990.7	4983.1 (1)		
$\nu_{1}$ + $\nu_{3}$	4487.3	4480.6 (1)		
$\nu_{1}$ + $\nu_{4}$	3930.5	3931.4 (1)		
$\nu_{1}$ + $\nu_{5}$	7037.8	7034.3 (5)	7041.8 ${}^{d}$, 7050 ${}^{b}$	
$\nu_{1}$ + $\nu_{6}$	4879.9	4873.8 (1)	4827.49 ${}^{c}$	
$\nu_{2}$ + $\nu_{3}$	2258.6	2251.4 (1)		
$\nu_{2}$ + $\nu_{4}$	1736.0	1736.2 (1)		
$\nu_{2}$ + $\nu_{5}$	4989.4	4981.8 (1)	4982.57 ${}^{c}$	
$\nu_{2}$ + $\nu_{6}$	2663.8	2657.2 (5)		
$\nu_{3}$ + $\nu_{4}$	1191.2	1193.0 (1)		
$\nu_{3}$ + $\nu_{5}$	4484.9	4478.2 (1)	4487.27 ${}^{c}$	
$\nu_{3}$ + $\nu_{6}$	2141.1	2135.5 (1)		
$\nu_{4}$ + $\nu_{5}$	3926.6	3927.4 (1)		
$\nu_{4}$ + $\nu_{6}$	1576.0	1578.6 (1)		
$\nu_{5}$ + $\nu_{6}$	4876.7	4870.7 (1)		

*^a^* Previous theoretically-attributed overtone from [63]. *^b^* Previous gas-phase experimentally-attributed overtone from [65]. *^c^* Previous gas-phase experimentally-attributed two-quanta modes from [64]. *^d^* Previous gas-phase experimentally-attributed combination band and computationally-attributed overtone from [66]. This attribution is questioned herein. See text for discussion.

**Table 4 molecules-27-03200-t004:** Geometrical parameters and spectroscopic constants for HOSH compared to the previous theory and gas-phase experiment.

	Units	CcCRE	CcCR	F12-TZ	Prev. Theory	Prev. Expt.
R${}_{e}$(H${}_{1}$-O)	Å	0.95700	0.96012	0.96152	0.9601 ${}^{b}$	0.9606 ${}^{c}$
R${}_{e}$(O-S)	Å	1.65350	1.66051	1.66370	1.6614 ${}^{b}$	1.6616 ${}^{c}$
R${}_{e}$(S-H${}_{2}$)	Å	1.33758	1.34273	1.34423	1.3413 ${}^{b}$	1.3420 ${}^{c}$
${∠}_{e}$(H${}_{1}$-O-S)	${}^{\circ }$	107.53	107.21	107.20	107.0 ${}^{b}$	107.19 ${}^{c}$
${∠}_{e}$(O-S-H${}_{2}$)	${}^{\circ }$	98.38	98.45	98.43	98.6 ${}^{b}$	98.57 ${}^{c}$
$\tau_{e}$(H${}_{1}$O-S-H${}_{2}$)	${}^{\circ }$	91.58	91.31	91.42	91.3 ${}^{b}$	90.41 ${}^{c}$
$A_{e}$	MHz	205,070.3	203,373.2	202,852.8	203,624 ${}^{b}$	
$B_{e}$	MHz	15,546.3	15,420.7	15,362.9	15,406 ${}^{b}$	
$C_{e}$	MHz	15,116.1	14,995.6	14,940.2	14,985 ${}^{b}$	
R${}_{0}$(H${}_{1}$-O)	Å	0.95493	0.95805	0.95947		
R${}_{0}$(O-S)	Å	1.66203	1.66926	1.67248		
R${}_{0}$(S-H${}_{2}$)	Å	1.34726	1.35227	1.35379		
${∠}_{0}$(H${}_{1}$-O-S)	${}^{\circ }$	107.80	107.51	107.49		
${∠}_{0}$(O-S-H${}_{2}$)	${}^{\circ }$	98.44	98.50	98.49		
$A_{0}$	MHz	203,686.6	202,021.5	201,494.3	202,199 ${}^{b}$	202,069 ${}^{a}$
$B_{0}$	MHz	15,428.1	15,299.9	15,242.2	15,285 ${}^{b}$	15,282 ${}^{a}$
$C_{0}$	MHz	14,983.2	14,860.4	14,805.3	14,847 ${}^{b}$	14,840 ${}^{a}$
$A_{1}$	MHz	201,230.6	199,597.1	199,079.8	199,769.9 ${}^{d}$	199,532.6 ${}^{d}$
$B_{1}$	MHz	15,403.5	15,275.4	15,217.9	15,255.6 ${}^{d}$	15,260.0 ${}^{d}$
$C_{1}$	MHz	14,971.9	14,849.0	14,794.0	14,829.6 ${}^{d}$	14,833.2 ${}^{d}$
$A_{2}$	MHz	199,585.2	197,980.8	197,464.1		
$B_{2}$	MHz	15,447.6	15,318.6	15,260.6		
$C_{2}$	MHz	14,982.9	14,859.6	14,804.2		
$A_{3}$	MHz	206,750.4	205,046.1	204,484.5		
$B_{3}$	MHz	15,386.9	15,257.9	15,200.6		
$C_{3}$	MHz	14,986.0	14,861.5	14,806.5		
$A_{4}$	MHz	206,017.2	204,343.0	203,808.7		
$B_{4}$	MHz	15,414.9	15,286.0	15,228.1		
$C_{4}$	MHz	14,920.8	14,797.9	14,742.9		
$A_{5}$	MHz	203,499.1	201,823.1	201,296.5		
$B_{5}$	MHz	15,299.2	15,168.7	15,111.4		
$C_{5}$	MHz	14,855.8	14,730.7	14,675.9		
$A_{6}$	MHz	202,269.5	200,635.4	200,115.3		
$B_{6}$	MHz	15,384.1	15,255.2	15,197.5		
$C_{6}$	MHz	14,912.1	14,789.6	14,734.6		
$\Delta_{J}$	kHz	23.614	23.535	23.346		24.528463 ${}^{a}$
$\Delta_{K}$	MHz	5.604	5.495	5.473		5.989715 ${}^{a}$
$\Delta_{JK}$	kHz	388.724	385.632	382.477		0.3904340 (MHz) ${}^{a}$
$\delta_{J}$	Hz	682.725	678.344	671.200		
$\delta_{K}$	MHz	−0.811	−0.823	−0.816		
$\Phi_{J}$	mHz	−12.453	−14.415	−14.509		
$\Phi_{K}$	Hz	448.475	433.966	429.934		
$\Phi_{JK}$	Hz	−8.868	−9.067	−8.931		
$\Phi_{KJ}$	Hz	58.332	58.290	57.557		
$\phi_{j}$	μHz	484.538	407.691	394.947		
$\phi_{jk}$	mHz	−862.352	−710.680	−699.524		
$\phi_{k}$	kHz	−7.755	−7.988	−7.893		
$\mu_{a}$	D			1.41		
$\mu_{b}$	D			0.76		
$\mu_{c}$	D			0.05		
$\mu_{net}$	D				1.66 ${}^{b}$	

*^a^* Previous gas-phase ground state rotational constants and centrifugal distortion constants from [67]. *^b^* Previous theoretical geometrical parameters calculated at the CCSD(T,full)/cc-pwCVQZ level of theory, ground state rotational constants calculated at the CCSD(T, full)/cc-pwCVQZ level of theory with vibration rotation corrections from the CCSD(T)/cc-pV(T+d)Z level of theory, and theoretical dipole moment calculated at the CCSD(T, full)/cc-pwCVQZ level of theory from [68]. *^c^* Previous empirical equilibrium geometrical parameters from [69]. *^d^* Previous theoretical rotational constants at the CCSD(T)/cc-pVQZ level of theory with CCSD(T)/cc-pVTZ vibrational correction and experimental rotational constants from [70].

**Table 5 molecules-27-03200-t005:** Vibrational frequencies (cm${}^{-1}$), and IR intensities (km/mol) given in parentheses for HOSH compared to previous theory and gas-phase experiments.

Mode	Desc.	CcCRE	CcCR	F12-TZ	Prev. Theory	Prev. Gas-Phase	Prev. Ar Matrix
$\omega_{1}$	S${}_{1}$	3854.6	3825.8	3819.6	3829 (69) ${}^{a}$		
$\omega_{2}$	S${}_{3}$	2672.4	2662.2	2656.1	2649 (16) ${}^{a}$		
$\omega_{3}$	S${}_{4}$	1210.4	1210.9	1211.0	1228 (41) ${}^{a}$		
$\omega_{4}$	S${}_{5}$	1043.8	1033.7	1032.8	1029 (2) ${}^{a}$		
$\omega_{5}$	S${}_{2}$	794.1	786.0	784.7	777 (52) ${}^{a}$		
$\omega_{6}$	S${}_{6}$	487.1	479.8	479.6	490 (75) ${}^{a}$		
$\nu_{1}$	S${}_{1}$	3650.8	3628.9	3626.7 (67)	3646.5 ${}^{b}$	3625.6 ${}^{b}$	3608.3 ${}^{b}$
$\nu_{2}$	S${}_{3}$	2556.8	2547.0	2544.9 (8)	2533.3 ${}^{b}$	2538 ${}^{b}$	2550.1 ${}^{b}$
$\nu_{3}$	S${}_{4}$	1173.5	1174.8	1176.7 (36)	1183.5 ${}^{b}$		1175.7 ${}^{b}$
$\nu_{4}$	S${}_{5}$	1018.8	1009.3	1008.3 (2)	1006.6 ${}^{b}$		
$\nu_{5}$	S${}_{2}$	772.2	763.7	763.4 (47)	764.4 ${}^{b}$		762.5 ${}^{b}$
$\nu_{6}$	S${}_{6}$	441.9	438.0	447.4 (72)	448.1 ${}^{b}$		445.3 ${}^{b}$
ZPT		4951.8	4922.8	4920.2			

*^a^* Previous computed harmonic frequencies conducted at the CCSD(T)/cc-pV(T+d)Z level of theory from [68]. *^b^* Previous theory, conducted at the CCSD(T)/cc-pVTZ level of theory, gas-phase experiment, and Ar Matrix data from [73].

**Table 6 molecules-27-03200-t006:** Vibrational frequencies (cm${}^{-1}$), and IR intensities (km/mol) given in parentheses for two-quanta bands of HOSH compared to the previous gas-phase experiment.

Mode	CcCR	F12-TZ	Prev. Theory
$2\nu_{1}$	7075.6	7074.7 (5)	
$2\nu_{2}$	4986.7	4985.7 (1)	
$2\nu_{3}$	2330.8	2334.7 (4)	
$2\nu_{4}$	2010.7	2008.8 (1)	
$2\nu_{5}$	1524.8	1522.5 (1)	
$2\nu_{6}$	839.1	864.0 (3)	846.269 ${}^{a}$
$\nu_{1}$ + $\nu_{2}$	6174.8	6170.5 (1)	
$\nu_{1}$ + $\nu_{3}$	4781.8	4782.1 (1)	
$\nu_{1}$ + $\nu_{4}$	4637.1	4633.8 (1)	
$\nu_{1}$ + $\nu_{5}$	4394.2	4390.7 (1)	
$\nu_{1}$ + $\nu_{6}$	4063.4	4071.3 (1)	
$\nu_{2}$ + $\nu_{3}$	3721.6	3721.4 (1)	
$\nu_{2}$ + $\nu_{4}$	3537.0	3533.9 (1)	
$\nu_{2}$ + $\nu_{5}$	3315.6	3312.2 (1)	
$\nu_{2}$ + $\nu_{6}$	2988.3	2997.4 (1)	
$\nu_{3}$ + $\nu_{4}$	2178.7	2179.4 (1)	
$\nu_{3}$ + $\nu_{5}$	1935.8	1936.2 (1)	
$\nu_{3}$ + $\nu_{6}$	1612.0	1626.5 (1)	
$\nu_{4}$ + $\nu_{5}$	1768.6	1766.2 (1)	
$\nu_{4}$ + $\nu_{6}$	1448.0	1456.5 (1)	
$\nu_{5}$ + $\nu_{6}$	1202.3	1209.7 (1)	

*^a^* Previous theoretically-attributed overtone at the CCSD(T)/aug-cc-pV(Q+d)Z level of theory from [74].

**Table 7 molecules-27-03200-t007:** Geometrical parameters and spectroscopic constants for HSSH compared to the previous gas-phase experiment.

	Units	CcCR	F12-TZ	Prev. Theory	Prev. Expt.
r${}_{e}$(H-S)	Å	1.34066	1.34213	1.3395 *^e^*	1.3421 ${}^{c}$
r${}_{e}$(S-S)	Å	2.05016	2.05579	2.0503 *^e^*	2.0564 ${}^{c}$
${∠}_{e}$(H-S-S)	${}^{\circ }$	98.19	98.17	98.18 *^e^*	97.88 ${}^{c}$
$\tau_{e}$(H-S-S-H)	${}^{\circ }$	90.63	90.59	90.64 *^e^*	90.34 ${}^{c}$
$A_{e}$	MHz	147,873.5	147,533.6		
$B_{e}$	MHz	7023.8	6986.4		
$C_{e}$	MHz	7022.3	6984.9		
r${}_{0}$(H${}_{1}$-S${}_{1}$)	Å	1.34588	1.34733	1.34 ${}^{b}$	1.327${}^{c}$
r${}_{0}$(S${}_{1}$-S${}_{2}$)	Å	2.05883	2.06453	2.082 ${}^{b}$	2.055${}^{c}$
${∠}_{0}$(H${}_{1}$-S${}_{1}$-S${}_{2}$)	${}^{\circ }$	98.23	98.21	97.8 ${}^{b}$	91.33 ${}^{c}$
$A_{0}$	MHz	146,754.2	146,415.8		146,858.1473 ${}^{a}$
$B_{0}$	MHz	7010.6	6979.5		6970.42953 ${}^{a}$
$C_{0}$	MHz	6938.1	6894.1		6967.68832 ${}^{a}$
$A_{1}$	MHz	144,632.3	144,299.4		
$B_{1}$	MHz	7016.9	6985.7		
$C_{1}$	MHz	6945.2	6901.1		
$A_{2}$	MHz	149,577.8	156,292.4		
$B_{2}$	MHz	7001.9	7029.5		
$C_{2}$	MHz	6924.8	6880.8		
$A_{3}$	MHz	146,685.6	146,348.2		146,799.077 ${}^{g}$
$B_{3}$	MHz	6969.5	6938.4		6928.53044 ${}^{g}$
$C_{3}$	MHz	6898.1	6854.0		6926.67937 ${}^{g}$
$A_{4}$	MHz	145,893.9	145,559.8		
$B_{4}$	MHz	6982.3	6951.3		
$C_{4}$	MHz	6903.0	6859.0		
$A_{5}$	MHz	144,698.0	144,365.9	144,702.2 ${}^{f}$	
$B_{5}$	MHz	7016.2	6985.0	6977.638 ${}^{f}$	
$C_{5}$	MHz	6944.5	6900.5	6975.508 ${}^{f}$	
$A_{6}$	MHz	146,799.1	139,393.7		
$B_{6}$	MHz	6980.1	6890.5		
$C_{6}$	MHz	6914.5	6870.6		
$\Delta_{J}$	kHz	5.342	5.295		5.39849 ${}^{d}$
$\Delta_{K}$	MHz	2.268	2.261		2.42355 ${}^{d}$
$\Delta_{JK}$	kHz	85.417	84.400		85.5254 ${}^{d}$
$\delta_{J}$	Hz	−8.572	−8.395		
${}^{g}$ $\delta_{K}$	MHz	17.569	20.809		
$\Phi_{J}$	mHz	−1.211	−1.270		
$\Phi_{K}$	Hz	103.985	106.134		
$\Phi_{JK}$	Hz	5.649	6.891		
$\Phi_{KJ}$	Hz	−15.461	−19.644		
$\phi_{j}$	μHz	6.012	5.185		
$\phi_{jk}$	Hz	449.754	638.243		
μ	D		1.15	1.30 ${}^{b}$	

*^a^* Previous gas-phase ground state rotational constants from [78]. *^b^* Previous theoretical geometry and dipole moment at the CCSD(T)/cc-pVTZ level of theory from [79]. *^c^* Previous experimental structural data from [80]. *^d^* Previous experimental centrifugal distortion constants from [81]. *^e^* Previous theoretical geometrical parameters at the HF/VnZ(Q,5,6) + fc-CCSD(T)/VnZ(q,5) + CV/CCSD(T)/CVQZ level of theory with full-T,Q corrections and DPT2 corrections from [82]. *^f^* Previous experimental rotational constants from [83]. *^g^* Previous experimental rotational constants from [84].

**Table 8 molecules-27-03200-t008:** Vibrational frequencies (cm${}^{-1}$), and IR intensities (km/mol) given in parentheses for HSSH compared to the previous gas-phase experiment.

Mode	Desc.	CcCR	F12-TZ	Prev. Expt.
$\omega_{1}$ (*a*)	S${}_{2}$	2680.6	2674.8	
$\omega_{2}$ (*a*)	S${}_{3}$	908.1	907.9	
$\omega_{3}$ (*a*)	S${}_{1}$	529.9	528.0	
$\omega_{4}$ (*a*)	S${}_{4}$	443.7	441.1	
$\omega_{5}$ (*b*)	S${}_{5}$	2683.1	2677.2	
$\omega_{6}$ (*b*)	S${}_{6}$	907.3	907.7	
$\nu_{1}$ (*a*)	S${}_{2}$	2566.3	2563.5 (1)	2555.78 ${}^{b}$
$\nu_{2}$ (*a*)	S${}_{3}$	887.2	889.7 (1)	883 ${}^{a}$
$\nu_{3}$ (*a*)	S${}_{1}$	518.2	516.8 (1)	515.92230 ${}^{a}$
$\nu_{4}$ (*a*)	S${}_{4}$	405.8	417.6 (14)	416 ${}^{b}$
$\nu_{5}$ (*b*)	S${}_{5}$	2569.7	2566.2 (1)	2558.64 ${}^{b}$
$\nu_{6}$ (*b*)	S${}_{6}$	884.1	886.1 (2)	886 ${}^{b}$
ZPT		4020.2	4018.1	

*^a^* Previous gas-phase experimental values from [83]. *^b^* Previous gas-phase experimental values from [85].

**Table 9 molecules-27-03200-t009:** Vibrational frequencies (cm${}^{-1}$), and IR intensities (km/mol) given in parentheses for two-quanta bands of HSSH compared to the previous gas-phase experiment.

Mode	CcCR	F12-TZ	Prev. Expt.
$2\nu_{1}$	5085.8	5080.2 (1)	
$2\nu_{2}$	1762.5	1766.1 (1)	
$2\nu_{3}$	1031.8	1029.2 (1)	
$2\nu_{4}$	776.8	811.8 (1)	808.0 ${}^{a}$
$2\nu_{5}$	5078.4	5074.3 (1)	
$2\nu_{6}$	1760.7	1765.1 (1)	
$\nu_{1}$ + $\nu_{2}$	3447.1	3445.2 (1)	
$\nu_{1}$ + $\nu_{3}$	3087.9	3082.9 (1)	
$\nu_{1}$ + $\nu_{4}$	2977.8	2985.0 (1)	
$\nu_{1}$ + $\nu_{5}$	5028.9	5025.5 (1)	
$\nu_{1}$ + $\nu_{6}$	3447.0	3445.9 (1)	
$\nu_{2}$ + $\nu_{3}$	1398.6	1399.0 (1)	
$\nu_{2}$ + $\nu_{4}$	1290.7	1306.7 (1)	
$\nu_{2}$ + $\nu_{5}$	3443.0	3442.4 (1)	
$\nu_{2}$ + $\nu_{6}$	1763.5	1767.2 (1)	
$\nu_{3}$ + $\nu_{4}$	919.1	930.3 (1)	
$\nu_{3}$ + $\nu_{5}$	3084.3	3080.1 (1)	
$\nu_{3}$ + $\nu_{6}$	1398.0	1398.6 (1)	
$\nu_{4}$ + $\nu_{5}$	2973.8	2982.5 (1)	
$\nu_{4}$ + $\nu_{6}$	1281.3	1298.1 (1)	
$\nu_{5}$ + $\nu_{6}$	3444.2	3443.4 (1)	

*^a^* Previous gas-phase experimentally-attributed overtone from [78].

## Data Availability

All of the data supporting the conclusions of the present work are provided herein and in the Appendix A.

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
