# Peer review of "Spectral Signatures of Hydrogen Thioperoxide (HOSH) and Hydrogen Persulfide (HSSH): Possible Molecular Sulfur Sinks in the Dense ISM"

_molecules, 2022, doi:10.3390/molecules27103200_

Round 1

Reviewer 1 Report

The manuscript describes high-quality quantum chemical calculations of the ro-vibrational spectra of HOSH and HSSH, presented as potential sulfur sinks in the dense interstellar medium, and HOOH as a control. The article is well structured and clearly written; the results would be available to the experimental astrochemical/physical community, including those utilizing the new James Webb Space Telescope and the Atacama Large Millimeter/submillimeter Array for potential identification of the molecules, especially HOSH which is predicted to have the stronger signal.

I think the paper should be published once the following comments have been addressed adequately:

1) As an arm-chair astrochemist/physicist, I am skeptical that HOSH would actually be the sink.  S_8 comes to mind as being another potential candidate, for gas-phase molecules (like the case for C_60, doesn't Nature love symmetry?). The alternative mentioned in the manuscript of molecules adsorbed on dust particles also strikes me as more likely than gas-phase HOSH. But I may be wrong and the authors have presented valuable computational results that definitely should be published. 

2) Figure 3 is poor.  The choice of blue and black is unfortunate, particularly for the more diffuse parts.  I have difficulty telling what is what. Moreover, I don't see any line at 2538 cm^-1 as written in the text.  The Figure should be redone with clear indications of what lines (2538 and the lower J bands) one is meant to compare.

3) The referencing style is curious; they are numbered but ordered alphabetically by first author.  I suggest that straight numerical ordering would be much better.

Reviewer 2 Report

The search of the Sulfur-containing species similar to hydrogen peroxide in the dense interstellar medium is an actual task since they avoid astrophysical observations so far. Spectral detection of these HOSH and HSSH molecules requires accurate rovibrational constants in advance. They were obtained in the MS by ab initio quantum chemical computations through the quartic force field (QFF). 

The search of the anharmonic fundamental vibrational frequencies in such  four-atomic molecules was performed by the coupled cluster theory at the singles, doubles, and perturbative triples level CCSD(T) within the explicitly correlated F-12b formalism. This is quite sophisticated modern ab initio method, which provides rovibronic accuracy close to high-resolution spectroscopy. The core-electron correlation and scalar relativity corrections were accounted with the basis set limit extrapolation to crown all the conceivable accuracy. 

A big number of single-point energy calculations for every displacement in the QFF method were computed. Many other technical details of Hessian fitting were solved in the framework of the SPECTRO program to compute spectroscopic constants. The MS represents perfect theoretical study with interesting new results and deserves publication without any doubts. Some optional recommendation for minor revision are below.

Why the important anharmonic infrared intensities are calculated at rather simple MP2/aug-cc-pVDZ level of theory? At least the higher basis set could be used in order to provide comparable accuracy through-out the consistent whole study. A short comment is necessary as well as a new reference on the recent study of unharmonicity [R. R. Valiev at el. First-principles calculations of anharmonic and deuteration effects on photophysical properties of polyacenes and porphyrinoids. Physical Chemistry Chemical Physics 22 (39), 22314-22323 (2020).

Several two-quanta vibrational overtones and combination bands for HOOH 
have not been reported so far in the current literature. These new findings should be mentioned in conclusions. The relatively flat HOOH potential of the torsional motion represents a big problem for rotational constants and for anharmonic vibrational frequencies ν4. The new results are important and deserve more attention. New predictions for the NIR region are also quite important according to new experimental facilities in overtones and two-quanta detection. This message needs to be stresses for experimentalists. The great IR intensities for the HOSH molecule are evident because of high polarity of its chemical bonds. Not only the higher dipole moment but the peculiar S-O bond polarity is responsible for intense ν6 torsion with 72 km/mole. The vibrationally excited rotational constants represent another new findings which could be stressed.
